# Negative Symptoms and Behavioral Alterations Associated with Dorsolateral Prefrontal Syndrome in Patients with Schizophrenia

**DOI:** 10.3390/jcm10153417

**Published:** 2021-07-31

**Authors:** Pamela Ruiz-Castañeda, María Teresa Daza-González, Encarnación Santiago-Molina

**Affiliations:** 1Neuropsychological Evaluation and Rehabilitation Center (CERNEP), University of Almeria, Carretera de Sacramento, s/n, La Cañada de San Urbano, 04120 Almeria, Spain; prc750@ual.es; 2Department of Psychology, University of Almeria, Carretera de Sacramento, s/n, La Cañada de San Urbano, 04120 Almeria, Spain; 3Mental Health Hospitalization Unit of Torrecárdenas Hospital, Calle Hermandad de Donantes de Sangre, s/n, 04009 Almería, Spain; esantiag@ual.es

**Keywords:** schizophrenia, negative symptoms, fronto-subcortical syndromes, social functioning

## Abstract

The present study had three main aims: (1) to explore the possible relationships between the two dimensions of negative symptoms (NS) with the three frontal behavioral syndromes (dorsolateral, orbitofrontal and the anterior or mesial cingulate circuit) in patients with schizophrenia; (2) to determine the influence of sociodemographic and clinical variables on the severity of the two dimensions of NS (expressive deficits and disordered relationships/avolition); and (3) to explore the possible relationships between the two dimensions of NS and social functioning. We evaluated a group of 33 patients with schizophrenia with a predominance of NS using the self-reported version of the Frontal System Behavior scale. To quantify the severity of NS, the Assessment of Negative Symptoms (SANS) scale was used. The results revealed that the two dimensions of NS correlate positively with the behavioral syndrome of dorsolateral prefrontal origin. Regarding the influence of sociodemographic and clinical variables, in patients with a long evolution the NS of the expressive deficits dimension were less severe than in patients with a short evolution. A negative correlation was found between the severity of NS of the disordered relationships/avolition dimension and perceived social functioning. Our results show the importance of differentiating between the two dimensions of NS to characterize better their possible frontal etiology and impact on clinical course and social functioning.

## 1. Introduction

Negative symptoms (NS) in schizophrenia patients are defined as a decrease in normal functioning and behavior and have been mostly related to poor functional outcomes, producing a clear impact on family, social and occupational aspects [1]. Specifically, the five NS that are usually present in schizophrenia are: (1) flattening of affect, defined as a decrease in facial expression of emotions, eye contact and other movements that accompany speech; (2) alogia or decrease in verbal production or expressiveness; (3) avolition or apathy, defined as reduced initiation and persistence of goal-directed activity due to decreased motivation; (4) anhedonia or inability to experience pleasure from positive stimuli; and (5) asociality, understood as a reduction in social activity and a reduced interest or desire for establishing relationships with others [2].

Due to the impact of these symptoms on the patient, the study of these has increased considerably. Thus, some authors, through factor analysis studies, have reported the existence of two different factors that cover all the variability of NS. Specifically, this approach proposes the presence of two different dimensions of NS: a dimension of expressive deficits (also referred to as a decrease in emotional expression or expression factor), which includes the symptoms of flattened affect and alogia, and a second dimension that constitutes the factor of disordered relationships (also referred to as avolition-apathy or experiential factor). From now on, these will be referred to as disordered relationships/avolition which include the symptoms of apathy/avolition and anhedonia/asociality [3,4,5,6,7,8].

Recently, some studies have tried to identify whether these two dimensions can be distinguished from each other in terms of aspects such as work, social, demographic factors and clinical features [9,10,11]. For example, Llerena et al. [12] explored whether the two dimensions of NS were differentially related to work outcomes and found that the NS of disordered relationships/avolition, but not expressive deficits, were associated with fewer hours worked, poorer outcomes, lower wages and a lower probability of obtaining employment.

Rocca et al. [13] examined the functional outcomes in real life of a sample of outpatients, finding that disordered relationships/avolition was a more significant predictor of social activity, interpersonal relationships and a greater likelihood of being single, in addition to having greater stability in social symptoms. Likewise, Strauss et al. [14] report that those patients with symptoms of the disordered relationships/avolition dimension have been associated with worse social functioning and greater deficits in social cognition. This association is important, because social functioning is a key factor for the maintenance of patients in the community and constitutes a powerful predictor of the evolution of the disease. Furthermore, social functioning is a significant predictor of whether an individual may develop psychosis [15].

In this sense, different NS related to reduced motivation have been directly associated with poor social functioning. For example, for authors like Horan et al. [16] anhedonia is a fundamental factor underlying the disabling social isolation and emotional deterioration that result in schizophrenia, being directly related to reduced motivation to participate in social activities. Similarly, Schlosser et al. [17] reported a greater association between the symptoms of disordered relationships/avolition with social functioning. In their study they found that these symptoms were more predictive of social functioning compared to expressive deficits, suggesting that abulia and anhedonia are more important for determining the level of social functioning than flattening of affect and alogia.

Regarding sociodemographic variables, expressive deficit symptoms have been mostly related to the male gender, a lower age at onset of the disease [10,18] and have been negatively correlated with years of schooling [18]. Expressive deficits have also been related to a more impaired neuropsychological functioning, particularly deficits in executive functions and working memory [9,18], and more variable symptoms throughout the course of the disease [19,20]. Similarly, Gur et al. [21] reported a poorer quality of life in patients with these symptoms. However, Ergül and Üçok [18] also note that this dimension is related chiefly to asymptomatic remission after the first episode of the disease.

On the other hand, another important aspect to analyze regarding the two dimensions of NS is whether they are related to behavioral alterations associated with the prefrontal cortex. The prefrontal cortex (PFC) acts as a mediator in the specific functions carried out by the other cortical and subcortical structures. In this regard, Alexander et al. [22] identified a highly organized system of circuits that link portions of the frontal cortex with the basal ganglia and the thalamus.

At present, three fronto-subcortical circuits have been described that could be related to psychosis: (1) the dorsolateral circuit, which has its origins in the dorsolateral PFC with projections to the caudate nucleus (dorsolateral), globus pallidus (lateral dorsomedial) and thalamus (anterior ventral and medial dorsal); (2) the orbitofrontal circuit, which originates in the lateral orbital cortex with projections to the caudate (ventromedial) nucleus, globus pallidus (medial dorsomedial) and thalamus (ventral anterior and medial dorsal); and (3) the anterior cingulate or mesial circuit, which includes the anterior cingulate cortex, nucleus accumbens, globus pallidus (rostrolateral) and medial dorsal thalamus [23].

The interruption or failure of any of the structures of these circuits or their interconnections can produce, according to the affected area, a series of behavioral alterations [24], thus generating one of the following syndromes: the dorsolateral syndrome, the orbitofrontal syndrome or the anterior cingulate syndrome [25].

The dorsolateral syndrome is related to alterations in the most complex cognitive processes. The primary manifestation of failure in this circuit is evident in the following executive deficits: difficulties in planning, abstraction, working memory, fluency, mental flexibility, generation of hypotheses and working strategies, seriation and sequencing [26].

The orbitofrontal syndrome has been related to the regulation of affect and social emotions and behaviors, influencing behavioral disinhibition and emotional lability, along with decision-making based on affective states [26]. Alterations in the orbitofrontal circuit have also been related to the onset of uninhibited and self-centered behaviors and sometimes manic and euphoric states. The patient shows hyperactive but unproductive behavior, with emotionality ranging from euphoria to irritability, as well as a deficit in impulse control. An alteration of this circuit appears to disconnect the frontal monitoring systems from the limbic input [25].

The anterior cingulate syndrome has been related to apathy and its main manifestations are observed at the level of a deficit in emotional responses. Patients do not show reactivity to emotional stimuli and usually have poor initiation skills [27].

If we consider the behavioral and emotional alterations of each of these three syndromes that emerge as a result of an alteration of the fronto-subcortical circuits, essential similarities can be observed with the manifestations of NS described in schizophrenic patients [28,29,30,31,32,33]. These similarities lead us to speculate whether the dimensions of NS (expressive deficits and disordered relationships/avolition) could be specifically related to any of the behavioral alterations that have been linked to fronto-subcortical syndromes.

This latter question is relevant since it could help us to more precisely distinguish both the characteristics and pathological mechanisms presented by each dimension. Likewise, given the symptomatic heterogeneity of schizophrenia, it could help to inform the development of more targeted and effective interventions for cases of schizophrenia with a predominance of NS. However, to our knowledge, no studies have explored the possible relationships between the two dimensions of NS (expressive deficits and disordered relationships/avolition) and the behavioral alterations associated with the three frontal behavioral syndromes.

Thus, the present study had three main objectives. First, we aimed to explore in a sample of patients with schizophrenia with a predominance of NS the possible relationships between the two dimensions of NS (expressive deficits and disordered relationships/avolition) and the three frontal behavioral syndromes (dorsolateral prefrontal, orbitofrontal and anterior cingulate or mesial). Second, we set out to study the influence of sociodemographic (age, gender and years of education) and clinical variables (duration of illness, pharmacological treatment and clinical setting to which the patient belonged) on the severity of the two dimensions of NS (expressive deficits and disordered relationships/avolition). Finally, we aimed to study the relationship between the two dimensions of NS and the perception that patients have about their social functioning.

Considering the previous literature regarding our first objective, we expect that the dimension of expressive deficits is positively correlated with the dysexecutive behaviors associated with the dorsolateral syndrome. The dimension of disordered relationships/avolition would be expected to show a positive correlation with the behaviors associated with the orbitofrontal syndrome, because in the scientific literature the areas involved in this circuit have been related to changes in behavior, affecting interpersonal relationships and social interaction. In the same way, we expect apathic behaviors to be correlated with the anterior cingulate syndrome, since this dimension is characterized by a marked decrease in emotional expression and symptoms of apathy.

Regarding our second objective, we expect to find that men show a greater severity of symptoms of expressive deficits compared to women. With respect to the years of schooling, it would be expected that those patients with fewer years of schooling present greater severity of symptoms of expressive deficits than those with more years of schooling. Similarly, at the level of clinical variables, we expect that in this dimension of expressive deficits, patients with a longer duration of the disease differ from those with a shorter duration, since expressive deficits in the literature have been mainly associated with symptomatic remission after the first episode of the disease and with more variable symptoms throughout its evolution. Therefore, it would be expected that patients with a longer evolution could present a lower score for this symptomatology. Likewise, it is also expected that patients with a greater need for hospitalization differ significantly from those with outpatient care, because expressive deficits have been associated with poorer functional results over time.

With respect to social functioning, and according to data from the literature, we expect that those patients who score higher in the symptoms of disordered relationships/avolition will be those who have a lower score on the scale of social functioning (or those who show a worse perception of their social functioning).

## 2. Materials and Methods

### 2.1. Participants

In the present study, 33 participants (24 men and 9 women) aged between 18 and 57 years (M = 44.4, SD = 9.0) participated from the mental health area of the city hospital. The participants were recruited as part of a more extensive study exploring cognitive and emotional executive functions in a sample of patients with psychosis [34]. All participants included in the study had the following inclusion criteria: (1) definitive diagnosis of schizophrenia (paranoid), (2) a minimum of two years of confirmed diagnosis, (3) state compensated during the last months prior to the evaluation and (for this criterion, we had the collaboration of the patient’s psychiatrist who confirmed a psychopathological stability and active motivation to carry out the evaluation) (4) patients with a predominance of NS confirmed by a higher percentage score on the Scale for the Assessment of Negative Symptoms (SANS) than on the Scale for the Assessment of Positive Symptoms (SAPS).

The clinical and sociodemographic variables considered in this study can be seen in Table 1.

### 2.2. Assessment

#### 2.2.1. Negative Symptoms Assessment

The severity of NS was assessed using the Scale for the Assessment of Negative Symptoms (SANS) [35] and the Scale for the Assessment of Positive Symptoms (SAPS) [36].

*Scale for the Assessment of Negative Symptoms (SANS*) [35]. Made up of 30 items and 5 subscales that cover the following different negative symptoms: (1) flattening of affect, (2) alogia, (3) avolition-apathy, (4) anhedonia-asociality and (5) deterioration of attention. Rated on a scale from 0 (absence) to 5 (severe). Higher scores indicate greater presence and severity of NS. This scale also makes it possible to obtain a SN severity score from the dimension of expressive deficits (flattening of affect, allegiance) and another symptom severity score from the disordered relationships/withdrawal dimension (abulia-apathy, anhedonia-asociality). This scale has good test–retest reliability and validity [7].

*Scale for the Assessment of Positive Symptoms (SAPS)* [36]. Consists of 34 items and 4 subscales associated with the following main positive symptoms: (1) hallucinations, (2) delusions, (3) extravagant or strange behavior and (4) formal thought disorder. A score is obtained based on a scale from 0 (absence) to 5 (severe). A higher score indicates a greater presence and severity of positive symptoms. This scale has good test–retest reliability and validity [7].

#### 2.2.2. Behavioral Alterations Assessment

To identify and quantify the behavioral alterations associated with the three syndromes of frontal origin, we used the Spanish version [37] of the Frontal Systems Behavior Scale (FrSBe) [38]. This instrument consists of 46 items grouped into three independent subscales: executive dysfunction (17 items), disinhibition (15 items), and apathy (14 items). The scale provides a global measure of frontal deterioration but, in addition, depending on the partial scores obtained in each subscale, it discriminates between the three above-mentioned behavioral syndromes: dorsolateral prefrontal syndrome (executive dysfunction subscale), orbitofrontal syndrome (disinhibition subscale) and anterior cingulate syndrome (apathy subscale). This scale makes it possible to establish whether patients have clinically significant scores to indicate any of the three behavioral syndromes of frontal origin. By converting direct scores into standardized scores (T) according to the age, education and gender of the participant, three ranges of affectation can be obtained according to their cut-off point: no risk (<59 points); high risk or borderline (60 to 64); and clinically significant (>65). The FrSBe has two profile forms: the self-rating profile form and the family rating profile form. In the present study, only the self-rating profile form was used. The FrSBe has shown adequate construct validity to evaluate these three behavioral syndromes [37].

#### 2.2.3. Social Functioning Assessment

To measure social functioning we used the Spanish adaptation [39] of the short version of the Social Functioning Scale (SFS-HI) [40]. This scale was designed to evaluate patients’ perception of their social functioning. The scale has 15 items, providing a global score between 0 and 45 points. It has shown good reliability and validity [39]. We also used the domain of Social relationships of the quality of life questionnaire WHOQOL-BREF (World Health Organization Quality of Life) in its Spanish version [41]. This domain denotes an individual’s perception of personal relationships, social support and sexual activity.

### 2.3. Procedure

The patient’s attending professional (clinical psychologist or psychiatrist) administered the SANS and the SAPS. At the same time, the other scales (FrSBe, SFS-HI and WHOQOL-BREF) were applied by the researcher in a consultation. The scales were administered in a quiet office and took between 30 and 40 min to complete. These scales were completed by the investigator or by the patients, according to the preference of the patients themselves, always ensuring that they understood the instructions correctly and were sufficiently motivated to carry out the task. The procedure to establish the predominance of negative symptoms can be consulted in Ruiz-Castañeda et al. [34]. Once the sample of patients with a predominance of negative symptomatology was obtained, the scores of the two dimensions were obtained from the sum of the items of each dimension. The alogia/flattening of affect scores (items 1 to 15) were summed for expressive deficits, and apathy/asociality/anhedonia scores were summed (items 16 to 26) for disordered relationship/avolition.

### 2.4. Statistical Analysis

To study the possible relationships between the two dimensions of NS and the three frontal behavioral syndromes, Spearman correlation analyses were conducted between the two SANS scores (expressive deficits and disordered relationships/avolition) and the three FrSBe subscale scores (executive dysfunction, disinhibition and apathy).

To explore the possible influence of sociodemographic and clinical variables on the severity of the two dimensions of NS, the following analyses were performed. To test whether there were differences in the severity of the symptoms of the expressive deficits dimension between men and women, the Mann-Whitney U test was used. To explore the possible differences based on age, the patients were divided into two groups (<44.4 years and >44.4 years) and the scores obtained by both age groups in this dimension of NS were compared using the Mann–Whitney U test. To explore the influence of the years of education, the patients were divided into three groups: basic (<6), medium (7 and 12) and high (>12), and their scores in the expressive deficits dimension were compared with the Kruskal–Wallis test.

Regarding the clinical variables, for the pharmacological treatment the patients were divided into four groups (typical antipsychotics, atypical antipsychotics, typical and atypical antipsychotics or other medications not related to psychotic illness) and their scores were compared in this dimension of NS through the Kruskal–Wallis test. To explore the influence of the clinical device to which the patients belonged (hospital or outpatient treatment regimen) and the years of disease evolution (less than 11 years or more than 11 years), the comparison analyses were carried out through the Mann–Whitney U test. These same comparison analyses were also performed with the scores obtained in the disordered relationships/avolition dimension.

To analyze the relationship between social functioning and the two dimensions of NS, Spearman correlation analyses were performed.

## 3. Results

### 3.1. Negative Symptoms and Frontal Behavioral Syndromes

As expected, the expressive deficits dimension only showed a positive and significant correlation with the behavioral alterations associated with the dorsolateral prefrontal syndrome (executive dysfunction subscale) (*rho* = 0.5, *p* = 0.003). Patients that scored higher on the expressive deficits dimension also scored higher on the behavioral alterations associated with dysfunction in the dorsolateral prefrontal circuit. However, our predictions concerning the disordered relationships/avolition dimension were not supported, since a significant and positive correlation was observed with the behavioral alterations associated with the dorsolateral prefrontal syndrome (executive dysfunction subscale) (*rho* = 0.356, *p* = 0.042), but the correlation with the behavioral alterations associated with the orbitofrontal syndrome (disinhibition subscale) and the anterior cingulate syndrome (apathy subscale) was null (see Table 2).

These results are consistent with those obtained in the additional analyses. Based on the results obtained in the executive dysfunction subscale (dorsolateral prefrontal syndrome) and employing as a cut-off point the direct score considered clinically high (equal to or greater than 60), the patients were divided into two groups: (1) high in executive dysfunction (H-ED) and (2) low in executive dysfunction (L-ED). When comparing the scores of the two groups on the two NS dimensions, the H-ED group obtained significantly higher scores both in expressive deficits (*U* = −2.803, *p* = 0.003, *r* = 0.5) and disordered relationships/avolition (U = −2.013, *p* = 0.045, r = 0.35) (see Figure 1).

Finally, we considered it relevant to analyze the percentage of patients who presented a clinically significant score for each syndrome, since this could corroborate the behavioral abnormalities related to the frontal system in patients with a predominance of NS. Regarding the dorsolateral syndrome (executive dysfunction subscale), we found that 72.7% presented a clinically significant score; regarding the orbitofrontal syndrome (disinhibition subscale), 33.3% of the patients presented a clinically significant score; and concerning anterior cingulate syndrome (apathy subscale), 69.7% of patients obtained a clinically significant score (see Figure 2).

### 3.2. Influence of Sociodemographic and Clinical Variables on the Negative Symptoms

The sociodemographic and clinical characteristics of the sample are displayed in Table 1. Concerning the disordered relationships/avolition dimension, no significant differences were found according to age, gender, years of education, medication type or clinical setting (see Table 3). However, with regards to the dimension of expressive deficits (see Table 4), we found that patients with a short evolution, compared to those with a long evolution, showed significantly lower scores in this dimension of NS. No significant differences were found according to the rest of the demographic or clinical variables.

### 3.3. Social Functioning and Negative Symptoms

Regarding the two social functioning scores (Social Functioning Scale (SFS-HI) and the domain Social relationships of the quality of life questionnaire (WHOQOL-BREF), a significant correlation was found between the scores of the disordered relationships/avolition dimension and the scores of the *Social relationships* domain of the quality of life questionnaire (WHOQOL-BREF) (*rho* = −0.4, *p* = 0.03). The correlations with the remaining variables were null (see Table 5).

## 4. Discussion

The objectives of the present study were threefold. First, we aimed to explore the relationships between the two dimensions of NS symptoms with the behavioral alterations associated with three frontal behavioral syndromes: dorsolateral prefrontal syndrome (executive dysfunction subscale), orbitofrontal syndrome (disinhibition subscale) and anterior cingulate syndrome (apathy subscale). Our second goal was to analyze whether the two dimensions of NS (disordered relationships/avolition and expressive deficits) are influenced by clinical and sociodemographic variables. Finally, we aimed to explore the relationship between the two dimensions of NS and social functioning.

### 4.1. Negative Symptoms and Frontal Behavioral Syndromes

Regarding our first objective, we have used the FrSBE self-report questionnaire, the theoretical development of which was prompted by extensive previous research on frontal lobe circuits. The FrSBE is based on the model presented by Mega and Cummings [42] that reports clinically observable behaviors of the three fronto-subcortical circuits. In this model, executive dysfunction has been related to the dorsolateral prefrontal circuit, disinhibited behaviors have been associated with the orbital prefrontal circuit, and apathic behaviors have been related to the anterior cingulate circuit. Similarly, neuroimaging studies (voxel-based morphometry and voxel-based analysis of diffusion tensor images) by authors such as Kawada et al. [43] in patients with schizophrenia have associated the subscales of the FrSBE with different neuroanatomical correlates. Specifically, they have found an association between the subscale of executive dysfunction with the left and right dorsolateral prefrontal cortex. Tsujimoto et al. [44] have found that a higher score on the apathy subscale is related to the atrophy of different regions of the prefrontal cortex, including the orbitofrontal cortex, the frontal pole and the dorsolateral prefrontal cortex, this last area being related to apathy of a cognitive rather than an emotional type.

In our study, patients revealed a positive relationship between the dimension of expressive deficits and the behavioral alterations associated with the dorsolateral syndrome (executive dysfunction subscale). This finding suggests that this symptomatologic dimension of negative schizophrenia, characterized by a decrease in verbal production, particularly emotional expression, impoverished language and facial expressions (gestural and prosodic) could be associated with an alteration in the dorsolateral prefrontal circuit. Dorsolateral syndrome has been associated mainly with a deterioration of higher cognitive functions, particularly impairment of executive functions. Patients with this syndrome show significant deficits in solving complex problems, organizing, planning and carrying out goal-directed activities. These patients tend to appear inattentive, unmotivated, distracted and dependent on the environment at the behavioral level.

This finding is consistent with the results of previous studies that report the relationship between impoverished language, the flattening of affect, and the decrease of spontaneous movements with the deterioration of the dorsolateral prefrontal cortex, both at the level of schizophrenia and also in other pathologies such as depression or brain damage [45].

Regarding the dimension of disordered relationship/avolition (symptoms of apathy/avolition, anhedonia/asociality), we expected to find a relationship with the behavioral alterations associated with the anterior cingulate syndrome (apathy subscale), since the main affectation of this syndrome is observed at the level of emotional responses, reflecting a decrease in motivation and difficulties in initiating action, with apathy being the main behavioral symptom.

However, our results concerning this dimension did not support this notion since a positive correlation was observed with the alterations related to behaviors of the dorsolateral prefrontal syndrome (executive dysfunction subscale).

This correlation could provide a possible explanation for the symptom of apathy. Authors such as Zamboni et al. [46] state that this symptom, in general, can result from several different mechanisms and not only from altered processing of emotion and affect. Levy and Dubois [47] as well as Stuss and Knight [48] have categorized apathy into three subtypes: emotional apathy related to orbital brain areas; apathy of self-activation related to specific lesions of the basal ganglia and the limbic regions; and cognitive apathy or cognitive inertia related to the dorsolateral, ventrolateral and frontopolar area.

In our study, the correlation found with the behavioral alterations associated with the dorsolateral prefrontal syndrome (executive dysfunction subscale) could be taken to indicate that our patients, with predominantly negative schizophrenia, are more prone to apathy of a cognitive type, since this syndrome has been directly associated with apathy due to deficits in cognitive functions, expressly those of several executive functions necessary for goal-directed behavior (GDB), such as working memory, planning and cognitive flexibility.

Similarly, we expected a positive correlation between the dimension of disordered relationships/avolition with the orbitofrontal syndrome, since this syndrome has been related to behavioral alterations that affect adequate social functioning. In fact, authors such as Ohtani et al. [49], in their study using diffusion tensor imaging (DTI), analyzed the anomalies of the white matter within the connections of the medial orbitofrontal cortex, finding that these anomalies are related to more severe symptoms of anhedonia-asociality and avolition-apathy. However, in our study the disordered relationships/avolition symptoms were not related to the behaviors associated with the orbitofrontal syndrome. A possible explanation of our results could be found in that our sample of patients presents a lower percentage of uninhibited behaviors (33%) (see Figure 2), therefore speculatively the possibility of less affectation of the connections that make up the orbitofrontal circuit could be suggested.

These findings are relevant since it can be observed that regardless of the dimension of predominant NS, that is, expressive deficits (flattening of affect and alogia), or disordered relationships/avolition (symptoms of apathy/avolition, anhedonia/asociality), the behaviors associated with the dorsolateral prefrontal syndrome would be present. Authors such as Donohoe and Robertson [50] as well as Frith [51] raise the possibility that deficits in the dorsolateral area responsible for executive functioning could explain the NS of schizophrenia, where these NS are the consequence of a disorder of voluntary action, with impairments in self-initiated behavior, the selection of goal-directed responses and the inhibition of irrelevant responses.

In this sense, our findings on the association between dysexecutive behaviors (dorsolateral syndrome) and the two dimensions of NS, expressive deficits and disordered relationships/avolition, would be in line with neuroimaging studies that have reported the importance of the dorsolateral circuit in the severity of NS of schizophrenia [52,53,54]. Brady et al. [45] attempted to identify the correlates of the brain network with NS by means of the analysis of functional connectivity in the resting state, finding that the bilateral dorsolateral prefrontal cortex, especially the right area, significantly covary with NS in this study, observing how a disruption of the connectivity between the right dorsolateral prefrontal cortex and the midline of the cerebellum was strongly related to a greater severity of NS.

Similarly, Kawada et al. [43] examined in a sample of patients with schizophrenia the possible association between abnormalities of brain structures and their relationship with the behavioral deficits associated with the frontal system (dysexecutive, uninhibited and apathic behaviors). Although these authors found that patients showed a reduction in gray matter volume compared to controls in multiple frontal and temporal structures such as the bilateral dorsolateral prefrontal cortex, ventrolateral prefrontal cortex, medial prefrontal cortex, medial prefrontal cortex, anterior cingulate and orbitofrontal cortex, they only observed an association between dysexecutive behaviors and volume reduction in the dorsolateral prefrontal cortex; the other behaviors (apathic and uninhibited) did not show significant associations with any brain area.

Although behavioral alterations are not always parallel to brain or neuropsychological alterations, in fact authors such as Bechara et al. [55] have reported how damage to the orbital areas alters social behavior but preserves the patient’s cognitive ability to respond to conventional frontal lobe tests, such as the Wisconsin Card Sorting Test (WCST), this study by Kawada et al. [43] could suggest that dorsolateral prefrontal cortex pathology could be the neural basis for dysexecutive behaviors in patients with schizophrenia.

Regarding the clinical involvement of these patients in the three fronto-subcortical syndromes, we found that a large percentage of our patients presented a clinically significant score on the three syndromes, especially in the dorsolateral and anterior cingulate syndrome, with a lower proportion of patients with the orbitofrontal syndrome. The presence of these syndromes is understood as an indicator of behavioral abnormalities related to the frontal system [38]. Therefore, these results suggest that a large percentage of patients with schizophrenia with a predominance of NS present possible involvement of the three fronto-subcortical circuits.

### 4.2. Influence of Sociodemographic and Clinical Variables on Negative Symptoms

Regarding our second objective, only the expressive deficits dimension (flattening of affect and alogia) was influenced by the sociodemographic and clinical variables, although only in the case of disease duration, in which patients with a longer duration of the disease (>11 years) presented a significantly lower score on the symptoms than those patients with fewer years of disease (<11 years).

Although the course and evolution of NS to date is heterogeneous, some studies such as that of Ergül and Üçok [18] have found a relationship between symptomatic remission and the expressive deficits dimension score. This remission of symptoms can have different explanations, such as remission of positive symptoms or symptomatic improvement thanks to pharmacological treatment. Although in our study neither the type of therapy received (outpatient or in hospital) nor the pharmacological treatment were related to this symptomatology, the patients in our study regularly attend psychological and social therapies, therefore we could speculate that perhaps it is the dimension of disordered relationships that could benefit the most from this type of intervention.

### 4.3. Relationship between Social and Functioning Negative Symptoms

Finally, regarding our third objective in terms of social functioning, we found a negative correlation between the disordered relationships/avolition (symptoms of apathy/avolition, anhedonia/asociality) dimension and the domain of Social relations of the quality of life questionnaire (WHOQOL-BREF) that assesses the perception of patients about their personal relationships, social support and sexual activity. In our study, those participants with higher scores on this dimension were those who also obtained lower scores on the Social relations domain. This finding suggests that patients with marked symptoms of apathy, anhedonia and asociality are those patients who also have a worse perception of their social functioning, specifically a low perception regarding the quality of their personal relationships and social support. Ultimately, our results confirm those of other studies that have found a link between disordered relationships/avolition and poorer social functioning and more significant deficits in personal relationships [56,57]. These results are relevant due to the importance of social functioning in schizophrenia, coming to be considered a characteristic element of the disorder and being a key factor for the maintenance of patients in the community, in addition to being considered a predictor of the evolution of patients [58].

Regarding the score on the Social Functioning Scale (SFS-HI), we did not find correlations with this scale and the two dimensions of NS; this could be since, unlike the Social relations domain that specifically analyzes personal relationships, we used the short version of the scale that provides a single general score of different aspects in addition to social functioning, such as isolation/integration, leisure, autonomy or employment/occupation.

## 5. Conclusions and Implications

In conclusion, the study of NS from a dimensional perspective has allowed us to carry out a more exhaustive analysis of these two possible subgroups of patients with difficulties mainly in expression or emotional relationships, which is important given the symptomatic heterogeneity shown by patients with schizophrenia. This approach, therefore, allows us to identify more homogeneous clinical subgroups within the broader diagnosis of the disease.

In this regard, the main conclusion that we can draw from our results is the possibility that the dimension of expressive deficits may be related to a better evolution of the disease or, as has been suggested in other investigations, with remission of symptoms. Second, this study allowed us to analyze possible relationships with syndromes of frontal origin; we specifically highlight the possible involvement of the dorsolateral syndrome in the two dimensions of NS along with the potential implications of this finding. Dorsolateral syndrome has primarily been related to impaired executive functioning and, consequently, to a higher incidence of functional problems, so it would be interesting in future research to analyze whether these two symptomatic dimensions present different patterns of executive performance.

Additionally, we found that a high percentage of our patients presented the three fronto-subcortical syndromes. Although these syndromes have been reported in other populations, such as those with sudden brain damage or dementia, to our knowledge, this is the first study that has explored the relationship between behavioral abnormalities related to the frontal system and NS of schizophrenia.

Finally, we have confirmed other reports in the literature showing a possible relationship between the disordered relationships/avolition dimension and more impaired social functioning.

Ultimately, these findings and the reports of the previous literature in other pathologies could be taken to indicate the importance of addressing NS and their frontal lobe dysfunction from a neuropsychological model based on the disconnection syndromes because the dysfunction of the frontal system can produce similar symptoms in different pathologies. Numerous studies show a high prevalence of several NS in various pathologies that affect the frontal cortex and subcortical structures, as is the case of fronto-cortical dementia, depression, dementia of the frontal lobe and acquired brain damage. Therefore, the approach of studying a set of symptoms present in different diseases could help to further clarify the causes and areas involved in such disorders.

At a therapeutic level, these results indicate the importance of including possible deficits in executive functions in neurocognitive rehabilitation programs. For instance, it would be useful to work with those functions that are affected in dorsolateral prefrontal syndrome, such as working memory, cognitive flexibility, planning, monitoring tasks and selecting behaviors to solve problems. Likewise, analyzing NS in this more specific way by focusing on its two dimensions also allows us, at a therapeutic level, to potentially guide the design of individualized treatment plans that are adapted to the needs of each patient.

In summary, an in-depth exploration of the cognitive deficits of patients with NS from a two-dimensional perspective could guide us towards better interventions that would improve the quality of life and functionality of the patients.

## 6. Limitations

Our findings should be interpreted in the context of various limitations. First, we have a reduced number of participants, which could have compromised the power of the study. Second, all the patients included in the study were clinically stable, which would not allow the results to be extended to patients with psychotic decompensation or those in acute stages of the disease. Third, this was a cross-sectional study. Considering the characteristics of the evolution of schizophrenia, the time course during which the patient was performing the evaluation does not allow us to analyze the possible changes that they may have undergone throughout the disease. Therefore, the results of our sample could only be interpreted according to the disease duration (in years). Fourth, our study did not employ physiological or brain neuroimaging measures that would allow us to analyze specific patterns of each dimension of NS. Finally, regarding the clinical variable of pharmacological treatment, the sample has not been divided according to the calculation of an estimate based on chlorpromazine equivalents.

## Figures and Tables

**Figure 1 jcm-10-03417-f001:**
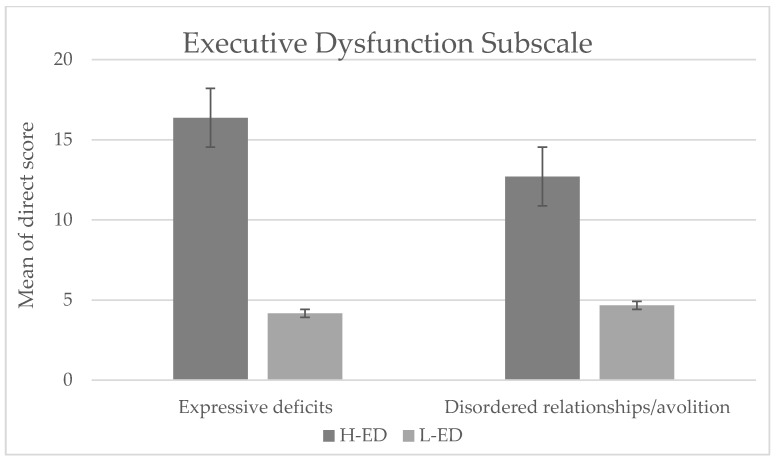
Mean direct score obtained on the two dimensions of NS (disordered relationships/avolition and expressive deficits) by the group of patients with high (H-ED) and low (L-ED) executive dysfunction.

**Figure 2 jcm-10-03417-f002:**
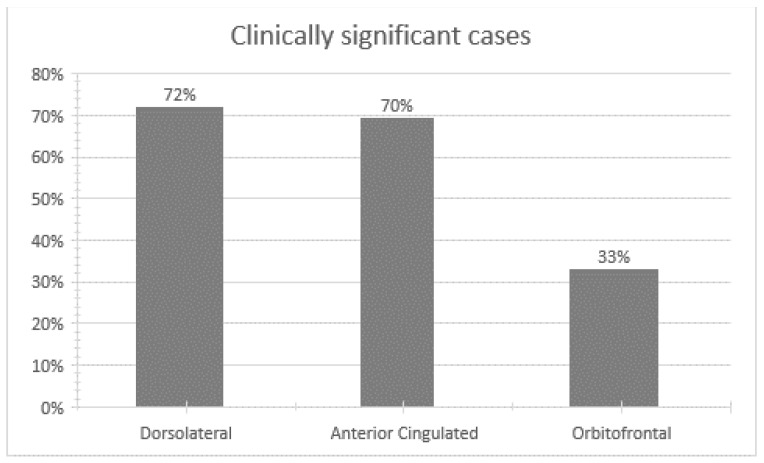
Percentage of clinically significant cases in fronto-subcortical syndromes in patients with negative symptoms.

**Table 1 jcm-10-03417-t001:** Sociodemographic and clinical variables of the patient sample.

Variables	Patients *N* = 33
	f (%)
**Sociodemographic**	
Age	
<44.4	15 (45.5)
>44.4	18 (54.5)
Gender	
Male	24 (72.7)
Female	9 (27.3)
Schooling (years)	
Basic (<6)	17 (51.5)
Medium (7 and 12)	9 (27.3)
High (>12)	7 (21.2)
**Clinical**	
Years of evolution of the disease	
Short evolution (less than 11 years)	16 (48.5)
Long evolution (more than 11 years)	17 (51.5)
Clinical setting	
In hospital	18 (54.5)
Outpatient	15 (45.5)
Pharmacological treatment	
Typical antipsychotics	4 (12.1)
Atypical antipsychotics	18 (54.5)
Typical and atypical antipsychotics	3 (9.1)
Other medications	8 (24.2)

**Table 2 jcm-10-03417-t002:** Correlations between the scores on the two dimensions of negative symptoms (disordered relationships/avolition and expressive deficits) and the scores on the three frontal behavioral syndromes (dorsolateral, orbitofrontal and anterior cingulate).

	Dorsolateral (Executive Dysfunction)	Orbitofrontal (Disinhibition)	Anterior Cingulate (Apathy)
**Expressive deficits**	0.496 *	0.029	0.211
**Disordered** **relationships/avolition**	0.356 **	−0.022	0.314
**Dorsolateral** **(Executive dysfunction)**	1	0.359*	0.593 *
**Orbitofrontal** **(Disinhibition) **	0.359 *	1	0.516 **
**Anterior cingulate (Apathy)**	0.593 **	0.516**	1

* *p* < 0.05; ** *p* < 0.01.

**Table 3 jcm-10-03417-t003:** Influence of sociodemographic and clinical variables on the severity of NS of the disordered relationships/avolition.

		Disordered Relationships/Avolition			
		Mean	SD	Statistics	*p*
Age	<44.4	9.40	7.77	*U* = 101.5	0.229
	>44.4	12.8	8.26		
Gender	Male	12.42	8.66	*U* = 76.5	0.207
	Female	8.11	6.17		
Schooling (years)	Basic	12.82	8.67	*Χ**^2^* = 0.848	0.654
	Medium	9.56	7.74		
	High	9.57	7.36		
Pharmacological treatment	Typical	12.75	9.63	*Χ**^2^* = 0.113	0.945
	Atypical	11.83	8.57		
	Both	10.00	6.55		
	Others	9.63	7.94		
Clinical setting	In hospital	9.67	5.93	*U* = 114.5	0.464
	Outpatient	13.13	10.01		
Years of evolution	Short evolution	11.56	7.96	*U* = 129.0	0.817
	Long evolution	10.94	8.45		

**Table 4 jcm-10-03417-t004:** Influence of sociodemographic and clinical variables on the severity of NS of the expressive deficits dimension.

		Expressive Deficits			
		Mean	SD	Statistics	*p*
Age	<44.4	13.80	13.5	*U* = 119.5	0.580
	>44.4	14.44	10.5		
Gender	Male	14.54	12.88	*U* = 104.0	0.890
	Female	13.11	8.66		
Schooling (years)	Basic	16.88	13.42	*Χ**^2^* = 1.143	0.565
	Medium	12.00	9.39		
	High	10.29	9.75		
Pharmacological treatment	Typical	15.50	15.67	*Χ**^2^* = 4.038	0.133
	Atypical	12.78	10.02		
	Both	32.00	14.93		
	Others	9.88	7.56		
Clinical setting	In hospital	13.44	12.66	*U* = 119.0	0.580
	Outpatient	15.00	10.99		
Years of evolution	Short evolution	18.19	12.91	*U* = 77.5	0.034 *
	Long evolution	10.35	9.44		

* *p* < 0.05.

**Table 5 jcm-10-03417-t005:** Correlations between scores on the two dimensions of negative symptoms (disordered relationships/avolition and expressive deficits), and scores on Social Functioning (SFS-HI) and the domain *Social relationships* of the quality of life questionnaire (WHOQOL-BREF).

	Social Functioning (SFS-HI)	Social Relationships
**Expressive deficits**	0.043	0.005
**Disordered relationships**	−0.048	−0.372*

* *p* < 0.05.

## Data Availability

The raw data supporting the conclusions of this article will be made available by the authors, without undue reservation.

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
