# Peer review of "Negative Symptoms and Behavioral Alterations Associated with Dorsolateral Prefrontal Syndrome in Patients with Schizophrenia"

_jcm, 2021, doi:10.3390/jcm10153417_

Round 1

Reviewer 1 Report

The article aims to explore the association among NS of  schizophrenia and 3 frontal behavioral syndrome, as well as with clinical, sociodemographic and social cognitive variables. Although the article it is well written, I have some comments that I will address below:

- In the introduction, the authors describe in detail the 3 frontal behavioral syndromes and their anatomic correlates, but they assess these syndromes using a self-report instrument. I think the authors should address the relationship between the neuroimaging correlates and the self-report used in their study.

-Additionally, in the introduction, the authors review in detail previous literature regarding NS and frontal behavioral syndromes, supporting their first aim of the study, but not at all the literature on social functioning or socioeconomic and clinical variables and NS. This results in a lack of coherence between the introduction and the rest of the manuscripts.   

- The description of the second aims (page 3, line 138-142) is not very clear to me. The authors split the data in many groups, some of them not related to functioning (e.g. gender). Do the authors expect differences among all groups to be found in the expressive deficits? And why?

- Page 3, line 143-144: the authors expect a difference between the patients with shorter and greater disease duration, but they do not mention a direction of this difference (e.g. greater duration more impairments?). The authors should explain why they did /can not hypothesize a direction in this difference or, based on the previous literature, they should provide a directionality in their hypothesis.

- I was wondering why the authors did not hypothesize any relationship between NS and the Orbitofrontal syndrome, since they state in the description of this syndrome, that impairments in this syndrome relates to disturbed social functioning and regulation of affect, which are main characteristics of the NS.

- My main concern is the data analysis strategy used for this study: I do not understand why the authors did not test all their aims using a regression model, in which the NS would be predicted by all the other variables. This strategy would also allow the investigation of which variable is the most predictive for NS, when controlling for all the other variables included in the model. Additionally, it is not clear for me why the authors split the group in several subgroups in order to test their second hypothesis- this strategy decreases the statistical power (in some groups there were 7 or 9 patients).

-page 9, line 292- not sure what this analysis brings new to the article.

- I would suggest the inclusion of a full correlation table, because I believe it would be interesting to see the correlations between the frontal behavioral syndromes as well.

Some minor aspects:

  • In the introduction, some phrases are not clear and need revision in my opinion (page 2, lines 63-65 and 75-76)
  • When the authors introduced the aims of the study, they do not state clearly what is the population they will study (page 3 line 124-138). At line 141, page 3 the authors mention some “groups”, but they did not clearly explained what groups they will investigate.
  • Throughout the manuscript there are commas in the place of full stops (e.g results section page 6, line 292-298).
  • In the results section, page 6, the lines 278-279 should be deleted.
  • Figure 3, the values of percentages is not in the right format.

Reviewer 2 Report

The paper is well written and the methods are sound.

Introduction:

The introduction gives an appropriate overview over the topic and the aim of the study.

Methods:

The aspect of correlation the domains of negative symptoms with the frontal behavioral symptoms is an interesting approach for further differentiate among patients with NS.

It is a strength that all patients had a confirmed diagnosis of schizophrenia, although a comparison with first psychosis patients would be interesting.

Discussion:
Are there functional or structural investigations that undermine the association of dorsolateral prefrontal syndrome and the domains of negative symptoms? This would be interesting to mention. For orbitofrontal circuits an association to negative symptoms could be shown (Kirschner et al, Schizophrenia Bulletin 2021). On the other hand there is evidence that functional and structural correlates of the domains of negative symptoms may occur independently (Burrer et al, Schizophrenia research, 2020). It would be interesting to discuss if there is evidence of an association of clinically assessed frontal behavioral syndromes and corresponding functional and structural data. In particular in this study if the finding of an association of NS with dorsolateral prefrontal syndrome can be connected to structural or functional findings in dorsolateral prefrontal circuits.

The authors themselves mention the reduced number of participants as a limitation.
